# N$_2$ Fixation in *Trichodesmium* Does Not Require Spatial Segregation from Photosynthesis

Weicheng Luo,[a] Keisuke Inomura,[b] Han Zhang,[c] Ya-Wei Luo[a]

[a]State Key Laboratory of Marine Environmental Science and College of Ocean and Earth Sciences, Xiamen University, Xiamen, China
[b]Graduate School of Oceanography, University of Rhode Island, Narragansett, Rhode Island, USA
[c]Marine Genomics and Biotechnology Program, Institute of Marine Science and Technology, Shandong University, Qingdao, China

**ABSTRACT** The dominant marine filamentous N$_2$ fixer, *Trichodesmium*, conducts photosynthesis and N$_2$ fixation during the daytime. Because N$_2$ fixation is sensitive to O$_2$, some previous studies suggested that spatial segregation of N$_2$ fixation and photosynthesis is essential in *Trichodesmium*. However, this hypothesis conflicts with some observations where all the cells contain both photosystems and the N$_2$-fixing enzyme nitrogenase. Here, we construct a systematic model simulating *Trichodesmium* metabolism, showing that the hypothetical spatial segregation is probably useless in increasing the *Trichodesmium* growth and N$_2$ fixation, unless substances can efficiently transfer among cells with low loss to the environment. The model suggests that *Trichodesmium* accumulates fixed carbon in the morning and uses that in respiratory protection to reduce intracellular O$_2$ during the mid-daytime, when photosynthesis is downregulated, allowing the occurrence of N$_2$ fixation. A cell membrane barrier against O$_2$ and alternative non-O$_2$ evolving electron transfer also contribute to maintaining low intracellular O$_2$. Our study provides a mechanism enabling N$_2$ fixation despite the presence of photosynthesis across *Trichodesmium*.

**IMPORTANCE** The filamentous *Trichodesmium* is a globally prominent marine nitrogen fixer. A long-standing paradox is that the nitrogen-fixing enzyme nitrogenase is sensitive to oxygen, but *Trichodesmium* conducts both nitrogen fixation and oxygen-evolving photosynthesis during the daytime. Previous studies using immunoassays reported that nitrogenase was limited in some specialized *Trichodesmium* cells (termed diazocytes), suggesting the necessity of spatial segregation of nitrogen fixation and photosynthesis. However, attempts using other methods failed to find diazocytes in *Trichodesmium*, causing controversy on the existence of the spatial segregation. Here, our physiological model shows that *Trichodesmium* can maintain low intracellular O$_2$ in mid-daytime and achieve feasible nitrogen fixation and growth rates even without the spatial segregation, while the hypothetical spatial segregation might not be useful if substantial loss of substances to the environment occurs when they transfer among the *Trichodesmium* cells. Our study then suggests a possible mechanism by which *Trichodesmium* can survive without the spatial segregation.

**KEYWORDS** *Trichodesmium*, nitrogen fixation, oxygen, temporal segregation

**Ad Hoc Peer Reviewer** Noelle Held

Address correspondence to Ya-Wei Luo, ywluo@xmu.edu.cn.

The authors declare no conflict of interest.

**T**richodesmium sp. is a dominant contributor to marine microbial N$_2$ fixation (1–3), an essential process in marine ecology and biogeochemistry. N$_2$ fixation by *Trichodesmium* has been thought to be paradoxical, since it fixes N$_2$ and conducts O$_2$ evolving photosynthesis during the daytime (1), although nitrogenase, the enzyme catalyzing N$_2$ fixation, is highly sensitive to O$_2$ (4). One widely discussed hypothesis is that *Trichodesmium* may temporally segregate the two conflicting processes (5–10). Photosynthesis of *Trichodesmium* often peaks in the morning, while N$_2$ fixation mainly occurs at noon or in the afternoon, when the intracellular O$_2$ is low due to concurrent low photosynthesis and probably high respiration (5, 8, 9, 11, 12). This phenomenon of asynchrony in the peak timing of photosynthesis and N$_2$ fixation is commonly referred to as temporal segregation of the two processes,

although the two processes are not completely separated in time (5). The temporal segregation, however, is not always obvious in *Trichodesmium* (13–15), for unclear reasons.

In addition, it has been hypothesized that *Trichodesmium* segregates these two competing processes spatially (5, 7, 9, 10, 16). *Trichodesmium* exists as filamentous trichomes consisting of dozens to hundreds of cells (1), in which $N_2$ fixation may be allocated in specialized cell segments (termed diazocytes) (15) and thus be spatially segregated from photosynthesis (5, 9). The spatial segregation, if it exists, requires the transfer of substances among *Trichodesmium* cells, while it is unclear how it occurs (6). As supporting evidence, some studies have revealed that nitrogenase is only distributed in diazocytes (5, 17–19). However, contradictory results have also been reported in which nitrogenase is randomly distributed in some, or even all, *Trichodesmium* cells (20–22). $^{13}C$ and $^{15}N$ isotope measurements via nanometer-scale secondary ion mass spectrometry (NanoSIMS) also indicate that $N_2$ fixation of *Trichodesmium* might not be limited to specialized diazocytes (6). These results lead to controversy in the existence of spatial segregation between $N_2$ fixation and photosynthesis in *Trichodesmium*.

Multiple mechanisms have been found or proposed to be involved in the $O_2$ regulation of *Trichodesmium*. The reduced permeability of the *Trichodesmium* plasma membrane to $O_2$ can slow into-cell $O_2$ diffusion, which is possible considering that the membrane of Gram-negative *Trichodesmium* is surrounded by a cell envelope with multiple layers (23). A recent study also proposed that hopanoid lipid, a component of *Trichodesmium*'s membrane, may reduce the $O_2$ permeability (24).

Another mechanism for $O_2$ regulation is respiratory protection (RP). RP is active aerobic respiration of carbohydrates by diazotrophs to lower intracellular $O_2$ to protect nitrogenase, while the produced energy is lost as heat to the environment (5, 7, 25, 26). High RP might reduce the plastoquinone pool and send negative feedback to photosystem II (PSII), further lowering intracellular $O_2$ production in *Trichodesmium* (5).

Alternative electron transfer (AET), one of the photosynthetic electron transfer (PET) pathways, might also contribute to maintaining low intracellular $O_2$ (8, 26). Electrons produced from PSII via the decomposition of $H_2O$ transfer to ferredoxin (Fd) and terminally return to $H_2O$ by, e.g., the Mehler reaction, forming (pseudo)cyclic electron flows around photosystem I (PSI) and resulting in zero net $O_2$ production (27, 28). AET produces intracellular energy (ATP) but not the reducing agent NADPH (29). In marine diazotrophs, AET can be a complement to linear PET (LPET), in which ATP and NADPH are produced at a molar ratio (1.3:1) that is substantially lower than that (3:1) required by $N_2$ fixation (27, 28, 30–32). AET can therefore benefit $N_2$ fixation by providing ATP to energetically expensive $N_2$ fixation while not generating $O_2$ (8, 26, 33, 34).

Recently, the segregation of *Trichodesmium* $N_2$ fixation and photosynthesis has been studied using a physiological cell model (7). The model consists of coarse-grained metabolic fluxes resolving key metabolisms, such as $N_2$ fixation, respiration, and biomass synthesis, suggesting that combined mechanisms are essential in regulating intracellular $O_2$, including RP, low permeability of cell membranes, and temporal and spatial segregations of $N_2$ fixation and photosynthesis. However, the model predefined the temporal segregation of $N_2$ fixation and photosynthesis and thus did not test the possibility of *Trichodesmium*'s survival without spatial segregation. Exploring such a possibility is critical in reconciling conflicting observations in which nitrogenase is found in a small group of cells (5, 17–19) or all the cells (20–22).

In the present study, we constructed a systematic physiological model of a single trichome of *Trichodesmium*, tracking the fluxes of carbon, nitrogen, $O_2$, NADPH, and ATP through different intracellular pools and processes. An optimization method was applied to seek a model parameter set that maximizes the growth rate, which allows the model to self-organize its diurnal patterns of various physiological processes. Model experiments were conducted to quantitatively evaluate the importance and necessity of different strategies, including the temporal and spatial segregations of $N_2$ fixation and photosynthesis, for regulating the intracellular $O_2$ of *Trichodesmium* trichomes and accomplishing feasible $N_2$ fixation. The results show that the hypothetical spatial segregation can be useful but is not mandatory.

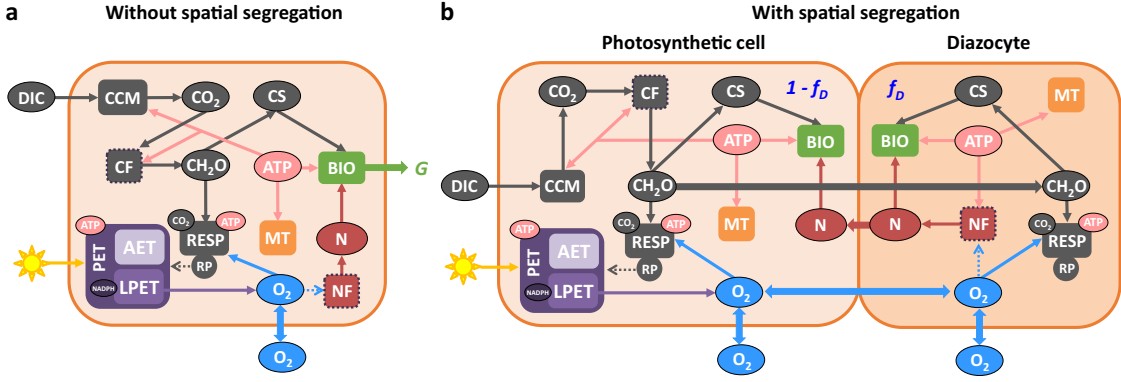

**FIG 1** Schematic diagram of our *Trichodesmium* trichome model. $N_2$ fixation and photosynthesis are not spatially segregated (a) or are spatially segregated (b). The model simulates the amount of biomass synthesized over a diel cycle and calculates a growth rate (*G*). In the model with spatial segregation (b), $N_2$ is fixed in diazocytes that consist of a fraction ($f_D$) of total cells, PET and carbon fixation occur in the remaining photosynthetic cells, and fixed carbohydrate produced in photosynthetic cells and fixed N produced in diazocytes are instantaneously transferred among the cells. For a clearer plot, arrows representing the production of ATP (by AET and LPET) and NADPH (by LPET), the consumption of NADPH (by both carbon and $N_2$ fixations [dashed frames]), and the transfer of ATP and NADPH from photosynthetic cells to diazocytes are omitted. ATP produced by RP is wasted and not counted. Refer to the text for detailed descriptions. Dark orange frame, cell membrane; oval blobs, biochemical pools; rectangles, metabolic processes; solid arrows, mass or energy fluxes; dashed arrows, inhibition effects of RP on PET (gray) or of $O_2$ on $N_2$ fixation (blue). CCM, $CO_2$-concentrating mechanism; CF, carbon fixation; NF, $N_2$ fixation; RP, respiratory protection; $CH_2O$, carbohydrates; CS, carbon skeleton; N, fixed nitrogen; MT, maintenance; BIO, biosynthesis.

## RESULTS

**General model framework.** The model (Fig. 1) estimates the growth of *Trichodesmium* trichome by simulating 12-h diurnal cycles of major intracellular processes involved in synthesizing organic carbon and fixing $N_2$. The production and the consequent allocation of intracellular ATP and/or NADPH partly determine the rates of these processes. The rate of $N_2$ fixation is also impacted by the temporal evolution of the intracellular $O_2$ level. The modeled $O_2$ inhibition on $N_2$ fixation rate uses a Michaelis-Menten equation (35), in which the half-saturation coefficient ($k_{O_2}^{NF} = 10^{-2}$ mol $O_2$ m$^{-3}$) is determined by fitting a modeled gross fixed C-to-N ratio of 49:1 with the observed value of 47:1 from an experiment with *Trichodesmium* (6) (see further discussion and sensitivity tests of $k_{O_2}^{NF}$ in the Discussion, below). The assimilated organic carbon and nitrogen accumulated at the end of the diurnal cycle are used to calculate a daily integrated growth rate. The spatial segregation of $N_2$ fixation and photosynthesis into segments of model trichome is also tested. Here, we briefly introduce our model framework, while more details are described in Materials and Methods.

We first introduce our model case in which photosynthesis and $N_2$ fixation are not spatially segregated (Fig. 1a). The model runs with diurnal variable light, which drives PET pathways, including LPET and AET. LPET produces $O_2$, ATP, and NADPH, while AET only produces ATP (28). The ratio of LPET and AET is dynamically adjusted to fulfill the relative requirements of ATP and NADPH by all the processes. The sole N source of the model, $N_2$ fixation, consumes ATP and NADPH (31, 32). $CO_2$ and $HCO_3^-$ are taken up (ATP is needed in $HCO_3^-$ uptake) (36) and then fixed to carbohydrates by consuming ATP and NADPH. Some carbohydrates are further synthesized to form carbon skeletons (37). ATP is also needed for cell maintenance (38). Our model prioritizes using ATP and NADPH for $N_2$ fixation over other processes, but $N_2$ fixation can only proceed under low intracellular $O_2$ levels (4).

The model implements RP by actively respiring carbohydrates to reduce the intracellular $O_2$ while wasting the produced ATP (5, 7, 25, 26). RP, as discussed above, inhibits PET (5) and consequently slows $O_2$ production. This intracellular production and consumption of $O_2$, as well as the cross-cell exchange of $O_2$, generate a dynamic level of intracellular $O_2$ and largely regulate the sometimes-observed diurnal patterns of $N_2$ fixation. Note that ATP and NADPH in the model are solely produced by LPET and AET during the daytime (39) and are instantaneously used (i.e., not stored). At the end of

**TABLE 1** Modeled growth and daily integrated carbon and $N_2$ fixation rates

| Model case | Growth rate (days$^{-1}$) | Carbon fixation rate (mol C [mol C]$^{-1}$ day$^{-1}$) | $N_2$ fixation rate (mol N [mol C]$^{-1}$ day$^{-1}$) |
|---|---|---|---|
| Without spatial segregation | 0.25 | 2.24 | 0.05 |
| With spatial segregation | 0.51 | 2.21 | 0.11 |

the daytime, the model calculates the amount of accumulated fixed carbon that needs to be respired to produce ATP, which subsequently supports maximal biosynthesis from the remaining fixed carbon and nitrogen.

The model case with spatially segregated photosynthesis and $N_2$ fixation is constructed by modifying the model without the spatial segregation, to separate the trichome into $N_2$-fixing (diazocytes) and photosynthetic cells (Fig. 1b). $N_2$ fixation is confined in diazocytes, which are set to make up 15% of total cells (2, 16), while LPET, AET, and carbon fixation only occur in the remaining photosynthetic cells. All the materials except $O_2$ are assumed to instantaneously and 100% efficiently transfer between diazocytes and photosynthetic cells and distribute evenly along the trichome (6), which is a best-case assumption for the growth of *Trichodesmium* with spatial segregation. Further evaluation and model experiments about this assumption are discussed later. Intracellular $O_2$ in diazocytes and in photosynthetic cells is simulated separately. A mixed layer of $O_2$ (see Fig. S1 in the supplemental material) is considered to form around the surface of the whole trichome, and the $O_2$ exchange among the mixed layer, the diazocytes and the photosynthetic cells, is calculated separately, following a scheme from Staal et al. (40).

**Growth rate and daily integrated carbon and $N_2$ fixation rates.** By optimizing model parameters, the model in which *Trichodesmium* trichome is not spatially segregated to diazocytes and photosynthetic cells achieves a maximal growth rate of 0.25 day$^{-1}$ (Table 1). The modeled growth rate falls within the general observed levels of *Trichodesmium* (0.1 to 0.4 day$^{-1}$) (13, 41–45). The modeled daily integrated fixed N (0.05 mol N per mol C per day) is coupled with the growth rate, using the Redfield ratio (46).

After incorporating the spatial segregation into the model, it reaches a much higher rate of 0.51 day$^{-1}$, which is mainly attributed to the elevated daily integrated $N_2$ fixation rate (Table 1). However, the daily integrated carbon (carbohydrate) fixation rate is nearly the same in the two model cases (Table 1), indicating that much more fixed carbon is respired or wasted in the model without spatial segregation. The high growth and $N_2$ fixation rates in the model without spatial segregation will be discussed later.

Nevertheless, *Trichodesmium* that does not spatially segregate photosynthesis and $N_2$ fixation can still grow mainly, because it fixes a large amount of carbon in the early period and then uses that carbon in RP during the mid-day, resulting in more $O_2$ consumption than is produced by photosynthesis and creating a low-$O_2$ window for $N_2$ fixation (further details are provided in the Discussion section, below).

**Temporal segregation of carbon and $N_2$ fixations.** The simulated carbon and $N_2$ fixations segregate temporally in both models with and without spatial segregation (Fig. 2a and b). These optimized results represent the patterns via which the model can reach the maximal growth rate and tentatively support the necessity of the temporal segregation between photosynthesis and $N_2$ fixation in *Trichodesmium*, more of which will be discussed later. The carbon fixation rate increases to its daily peak in the first 2 h, gradually decreases to approximately half of its maximum until noon, remains nearly constant (Fig. 2a) or increases to a second peak (Fig. 2b) for another 4 h, and then reduces to 0 at the end of the light period. $N_2$ fixation mainly occurs in the middle light period, when the carbon fixation rate is downregulated and a window of low intracellular $O_2$ emerges (Fig. 2c and d). Compared to the model without spatial segregation, the model spatially segregating photosynthesis and $N_2$ fixation has a wider low-$O_2$ window in diazocytes and a longer period of $N_2$ fixation (Fig. 2).

Our model generates a diurnal pattern of $N_2$ fixation with a single peak, which is consistent with most previous studies of single trichomes of *Trichodesmium* (5, 6, 8, 13, 39, 41, 44, 47). $N_2$ fixation in our model peaks during the later light period, which was observed in some of above studies (6, 13, 41, 47), although the exact peaking time of $N_2$ fixation varied substantially, probably due to different culture and physiological conditions.

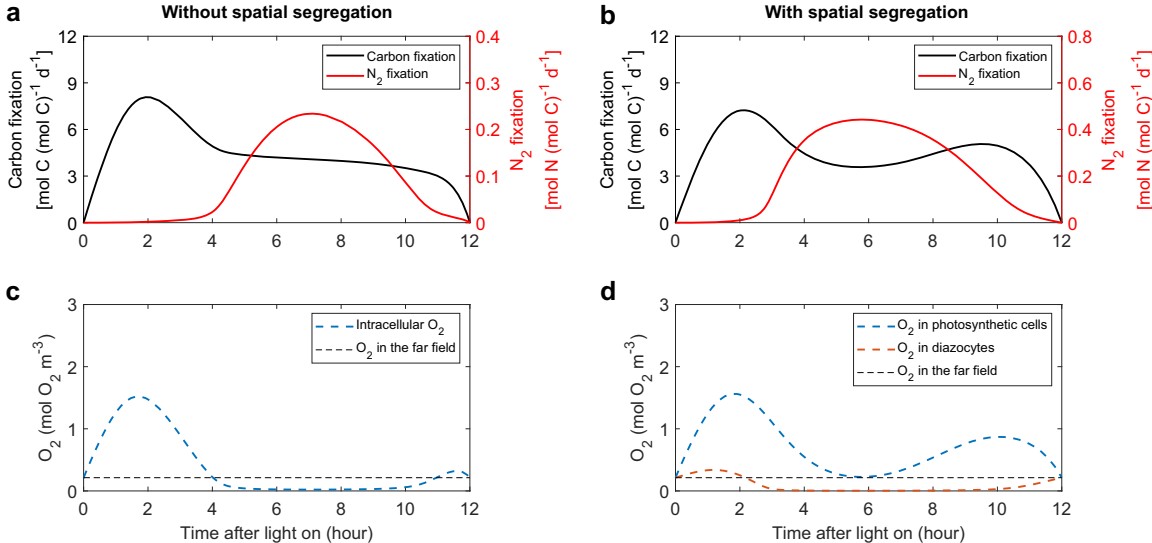

**FIG 2** Simulated rates of C and N₂ fixations and O₂ concentrations. The model runs both without (a and c) and with (b and d) a spatial segregation between carbon and N₂ fixation. The thin black dashed lines in panels c and d represent the ambient far-field O₂ concentrations set by the model.

**Dynamic changes of O₂ fluxes.** We first compared the intracellular O₂ budgets in the model without spatial segregation to those in the photosynthetic cells of the model with spatial segregation (Fig. 3a and b). In the early morning (0 to 4 h), a large amount of O₂ was quickly produced from photosynthesis, which is consistent with an observation that the gross O₂ evolution of *Trichodesmium* was high in the late morning or midday (48). The produced O₂ then either diffuses to the ambient environment or is respired in both cases. After that period, when O₂ production is moderate and N₂ fixation increases rapidly, the RP dominates the removal of intracellular O₂ in both cases. The low intracellular O₂ in turn leads to a physical influx of O₂ in the model without spatial segregation (Fig. 3a). Without N₂ fixation, the photosynthetic cells of the model with spatial segregation, however, allow a lower RP than that without spatial segregation, and meanwhile an intracellular O₂ concentration always higher than the extracellular level causes a continuous outflux of O₂ (Figs. 2d and 3b). For the diazocytes of the model with spatial segregation, no O₂ is produced inside, and there is only a relatively small influx of O₂ due to the small area of the interface (7); consequently, a low RP is enough to create a low-O₂ window in these cells (Figs. 2d and 3c).

In terms of the daily integrated O₂ budget, both the O₂ consumed by RP and its ratio to photosynthetic O₂ production in the model without spatial segregation are higher than those with spatial segregation (Fig. 3d). This is mainly because of the lowered RP requirement in both diazocytes and photosynthetic cells of the model without spatial segregation. Furthermore, with a higher intracellular O₂, the photosynthetic cells in the model with spatial segregation can diffuse a much larger amount of O₂ than that without spatial segregation (Figs. 2c and d and Fig. 3d).

**Carbon, ATP, and NADPH allocation.** Mainly owing to the much higher fraction of gross fixed carbon consumed by RP, much less (13%) fixed carbon is synthesized to biomass in the model without spatial segregation than that with spatial segregation (30%) (Fig. 4a). To supply ATP for biosynthesis at night, more fixed carbon is respired in the model with spatial segregation than that without spatial segregation because of the higher growth in the former (Table 1 and Fig. 4a). Compared to the model without spatial segregation, ATP production is higher in the model with spatial segregation, mainly because it is inhibited less by lower RP, with slightly more ATP produced by LEPT than by AET in both cases (Fig. 4c). Hence, the model with spatial segregation is capable of supporting higher energy consumption than that without spatial segregation (Fig. 4d). In both cases, most ATP (81% and 71% in models without and with spatial segregation, respectively) is consumed by carbon fixation, while much less ATP (4% and 8% in models without and with spatial segregation, respectively) is allocated

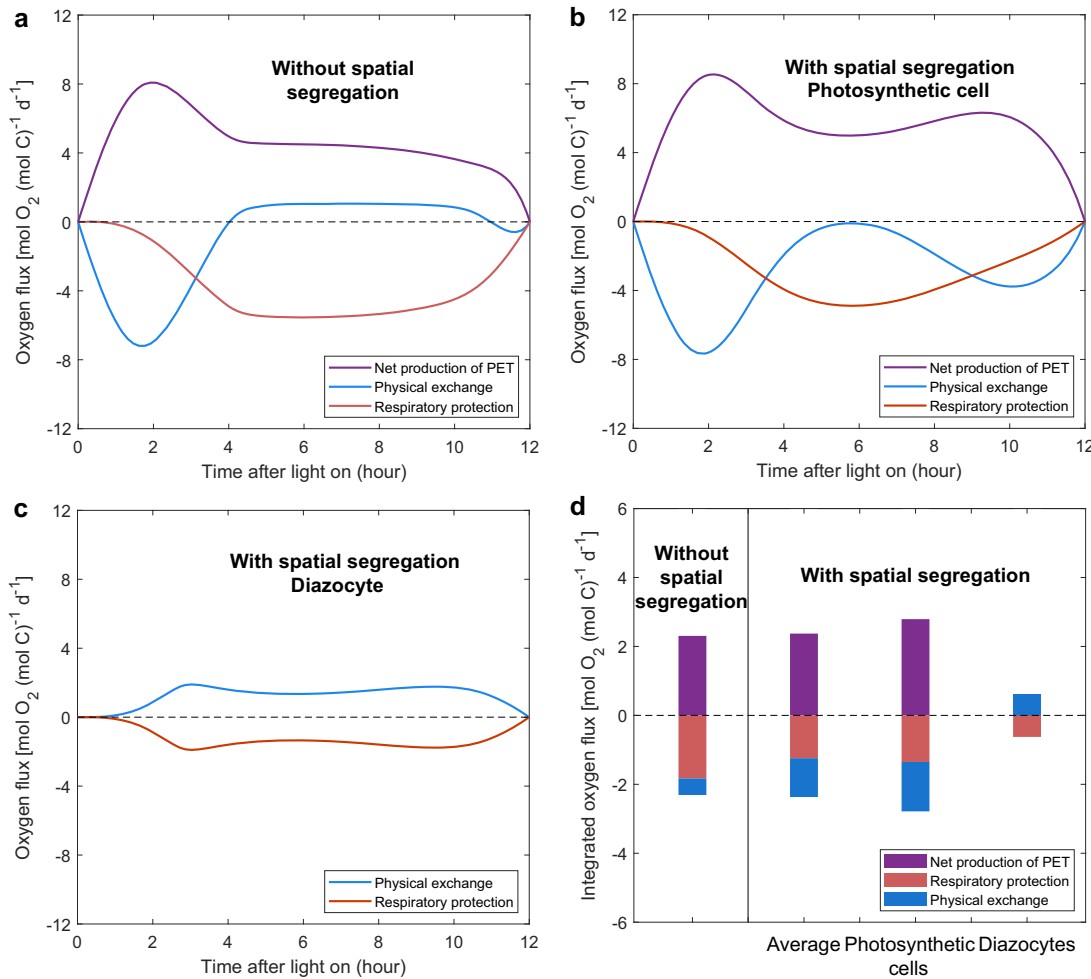

**FIG 3** Modeled intracellular $O_2$ fluxes. (a to c) Photosynthesis and $N_2$ fixation are either not spatially segregated (a) or spatially segregated to photosynthetic cells (b) and diazocytes (c). (d) Daily integrated $O_2$ fluxes in models with and without spatial segregation, with the results for the photosynthetic cells and diazocytes in the model with spatial segregation also shown. Positive and negative values represent gain and loss of intracellular $O_2$, respectively. $O_2$ fluxes in the photosynthetic cells and diazocytes (b to d) are normalized to their own respective biomass.

to $N_2$ fixation (Fig. 4d). The fraction of NADPH allocated to carbon fixation is even higher (97% and 93% in models without and with spatial segregation, respectively), with the remaining <10% of NADPH used by $N_2$ fixation, reflecting that carbon fixation requires a higher ratio of NADPH:ATP than $N_2$ fixation (Fig. 4b).

## DISCUSSION

**Formation of the temporal segregation.** Without representing the spatial segregation between photosynthesis and $N_2$ fixation in *Trichodesmium*, our model generates rhythms of carbon and $N_2$ fixations (Fig. 2) that are basically consistent with sometimes-observed rhythms (6). The modeled rhythms can be divided into four stages (Fig. 5).

In the first stage (hours 0 to 2), carbon is quickly fixed and accumulates, while $N_2$ is barely fixed due to high intracellular $O_2$ (see Fig. S2 in the supplemental material), resulting in an increasing ratio of particulate organic C to N, a phenomenon also found in culture experiments (49).

In the second stage (hours 2 to 4), the accumulation of carbon skeletons (see Fig. S2) increases the cellular demand for $N_2$ fixation, which in turn triggers RP (Fig. 3a). The elevated RP not only consumes more $O_2$ but also partly inhibits PET and $O_2$ production (Fig. 3a). These two effects, together with the diffusion of $O_2$ out of the cells, quickly reduce intracellular $O_2$ to a level lower than that in the environment (Fig. 2c).

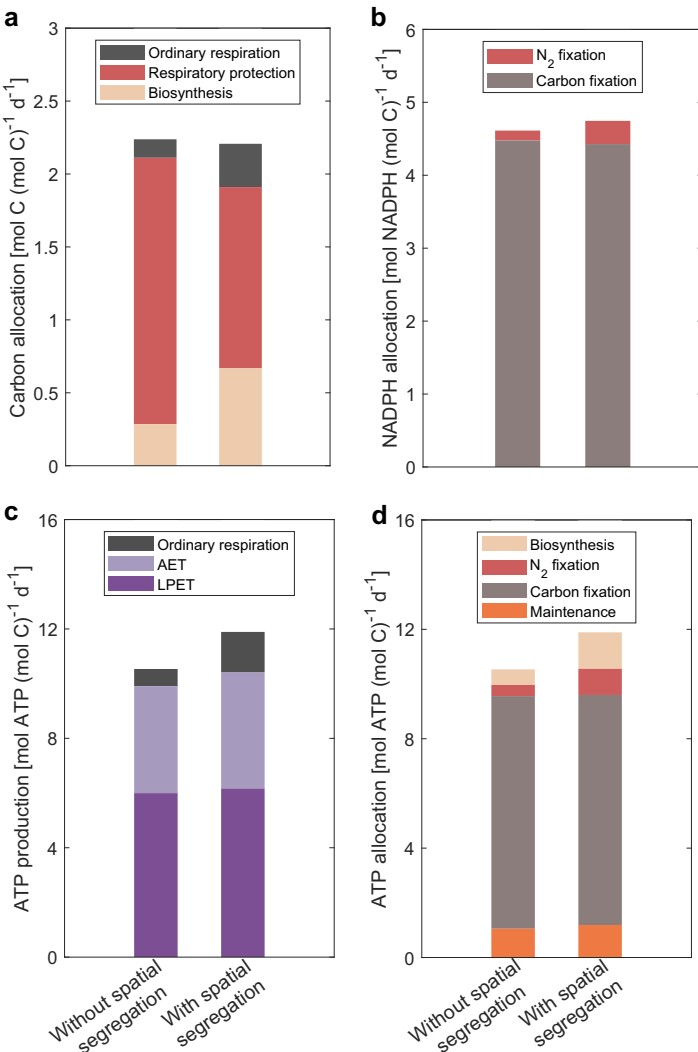

**FIG 4** Modeled daily integrated carbon, NADPH, and ATP fluxes. Gross fixed carbon (a) and NADPH (b) allocation and ATP production (c) and allocation (d) were integrated over a diel cycle in both models, without and with spatial segregation. Note that the ordinary respiration of fixed carbon is calculated at the end of the daytime for the amount of ATP needed to synthesize biomass during the night (see text for details).

In the third stage (hours 4 to 10.5), the majority of $N_2$ is fixed. To maintain a low intracellular $O_2$ for $N_2$ fixation (Fig. 2c), the RP is at its highest level (Fig. 3a) to consume not only all the $O_2$ that is photosynthetically produced at a moderate level (Fig. 2a) but also the $O_2$ influx from the environment. This consequently results in a net consumption of organic carbon (see Fig. S2). The results are consistent with the net $O_2$ consumption observed around a period of active $N_2$ fixation in a cultured *Trichodesmium* experiment (48). Therefore, adequate carbon must be fixed and stored in the first two stages before substantial $N_2$ fixation occurs, which is a reason for the necessity of the temporal segregation between carbon and $N_2$ fixations.

In the last stage (hours 10.5 to 12), the accumulation of fixed N (see Fig. S2) and downregulated PET because of weakened light (Fig. 3a) causes a decrease in $N_2$ fixation and in turn slows down RP (Fig. 3a). There is still a small amount of carbon fixed in this last stage (Fig. 2a).

Additional model experiments without the spatial segregation (see Text S1 and Fig. S3) show that the degree of temporal segregation between photosynthesis and $N_2$ fixation largely determines daily integrated $O_2$ production and RP and the ratio of net carbon to $N_2$ fixations. The model reaches a maximal growth rate at an intermediate level of the temporal segregation.

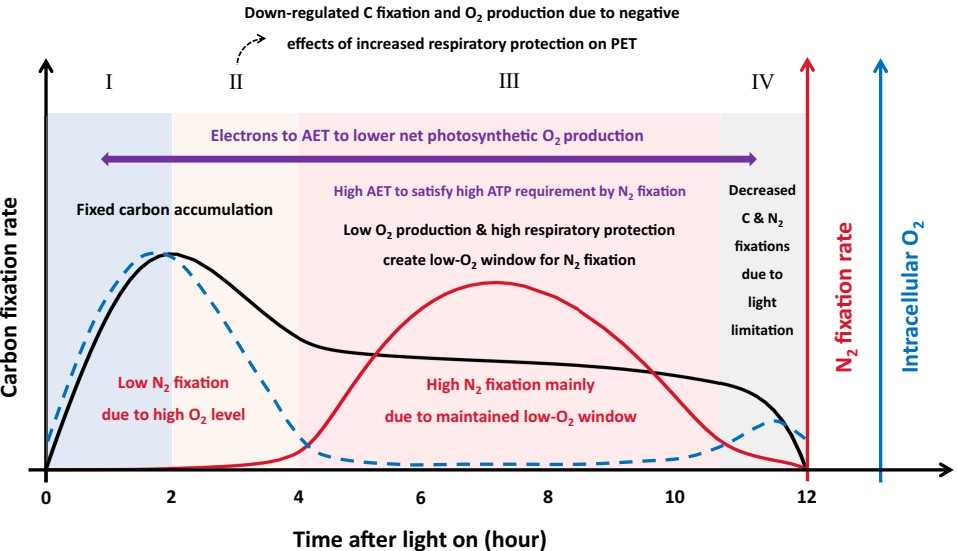

**FIG 5** Schematic diagram illustrating the temporal segregation between C and N$_2$ fixations in the model without spatial segregation. The modeled rhythms can be divided into four stages (I to IV). Black solid line, rate of carbon fixation; red solid line, rate of N$_2$ fixation; blue dashed line, intracellular O$_2$.

In summary, efficient carbon and N$_2$ fixations with dynamic regulation of intracellular O$_2$ and the requirement of sufficient accumulation of organic carbon before the period of high N$_2$ fixation are the two main reasons for the modeled temporal segregation between photosynthesis and N$_2$ fixation of *Trichodesmium*. Our model provides a scenario in which, even without spatially segregating N$_2$ fixation and photosynthesis, *Trichodesmium* can still grow at a moderate rate with the concurrence of the two processes.

Meanwhile, our model always produces the temporal segregation between N$_2$ fixation and photosynthesis, although some previous studies observed no temporal segregation in single trichomes of *Trichodesmium* (13, 14). The mechanism for how *Trichodesmium* grows without temporal segregation is certainly worthy of further investigations.

**Evaluation of the impacts from spatial segregation.** Meanwhile, the spatial segregation of photosynthesis and N$_2$ fixation in different cells can increase the modeled maximum growth rate by 104% (Table 1); this is mainly caused by the expanded low-O$_2$ window and the elevated N$_2$ fixation in diazocytes (Fig. 2) and the lowered consumption of fixed carbon in RP (Figs. 3 and 4a). This result, however, was obtained by assuming all the synthesized materials (except O$_2$) can freely and efficiently transfer between diazocytes and photosynthetic cells in the model. Although direct transfer of substances among cells has been suggested for some terrestrial filamentous N$_2$-fixing cyanobacteria, such as the channels found to connect cells in *Anabaena* (50, 51), such channels or other similar mechanisms have not been discovered for *Trichodesmium*. If the substances produced in certain cells of *Trichodesmium* have to be otherwise released to extracellular environment before they can be retaken by other *Trichodesmium* cells, the loss of the transferred substances to the environment would be unavoidable. By setting a lost fraction of the intercellularly transferred materials in the model with spatial segregation (see Materials and Methods), the growth rate decreases substantially, mainly caused by the loss of fixed N, and becomes even lower than that in the model without spatial segregation when the lost fraction is higher than 50% (see Fig. S4). A loss fraction lower than this level can be difficult to reach, considering the ocean environment is dynamic and other microorganisms inhabiting areas near *Trichodesmium* can also take up these substances. Our model experiments then suggest that the benefit that *Trichodesmium* can obtain from the spatial segregation is likely overwhelmed by the loss of substances during their transfer among cells.

**Intracellular O$_2$ management.** *Trichodesmium* also adopts another suite of combined intracellular O$_2$ management mechanisms to protect nitrogenase. Considering the

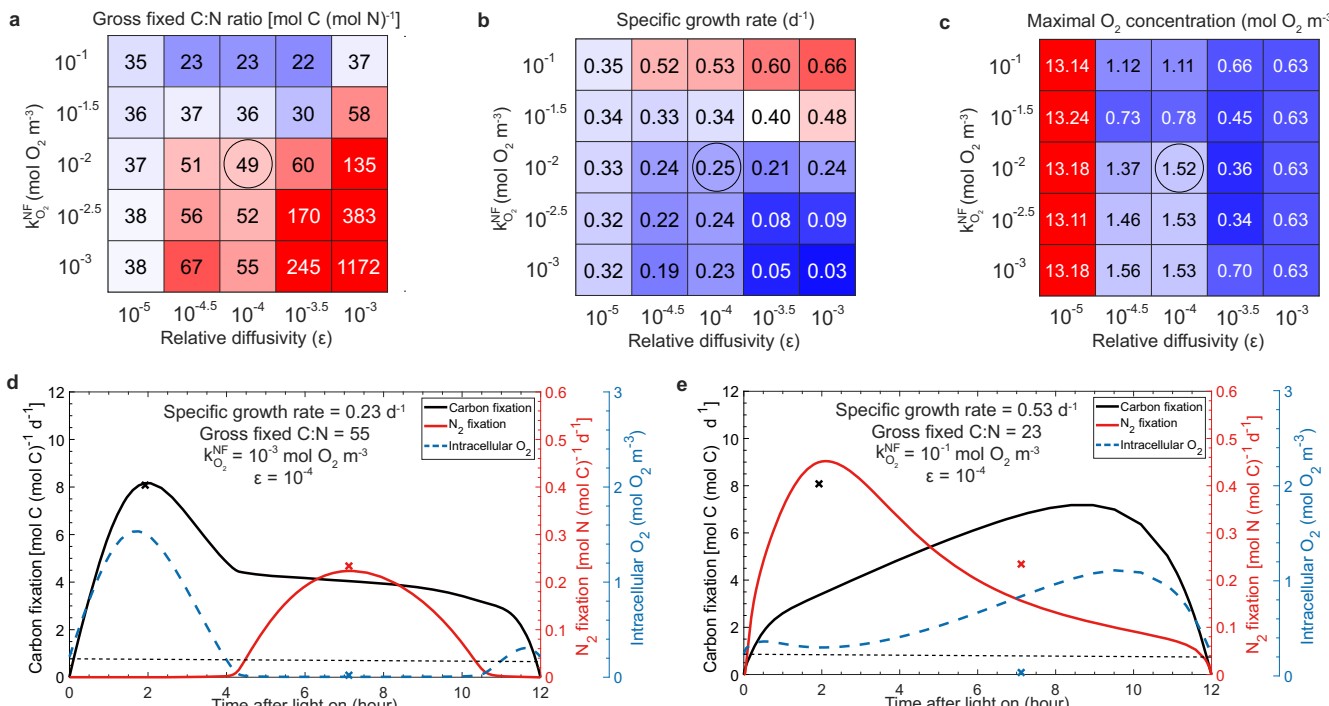

**FIG 6** Model sensitivity tests for two model parameters related to intracellular $O_2$ regulation. (a to c) The half-saturation coefficients of $O_2$ inhibition on $N_2$ fixation ($k_{O_2}^{NF}$) and the relative cross-membrane $O_2$ diffusivity to that in water ($\varepsilon$) were tested in the model without spatial segregation, showing a simulated ratio of gross C-to-N fixations (a), growth rates (b), and intracellular maximal $O_2$ concentration (c). Note that values in the black circles represent the model results simulated using default values for $k_{O_2}^{NF}$ and $\varepsilon$ (see text). (d and e) Optimized model results without spatial segregation using a low $k_{O_2}^{NF}$ ($10^{-3}$ mol $O_2$ m$^{-3}$) and a high $k_{O_2}^{NF}$ ($10^{-1}$ mol $O_2$ m$^{-3}$), respectively. For comparison, several key results of the standard model without spatial segregation using a default $k_{O_2}^{NF}$ ($1 \times 10^{-2}$ mol $O_2$ m$^{-3}$) (i.e., Fig. 2), including the maximal carbon fixation rate (black cross), the maximal $N_2$ fixation rate (red cross), and the minimal intracellular $O_2$ concentration (blue cross), are marked at their corresponding occurring times in panels d and e. The thin black dashed lines in panels d and e represent the ambient far-field $O_2$ concentrations predefined by the model.

analyses above, we limit our discussion in this section only to the model results without spatial segregation.

Our model results suggest that proper low cell permeability to $O_2$ is important to maintain the low-$O_2$ window for $N_2$ fixation, which is consistent with the conclusions of other studies (7, 48). The multilayer cell envelope of *Trichodesmium* makes the $O_2$ diffusivity across the cell membrane thousands of times lower than that in water (7, 48, 52). Our model experiment estimates an $O_2$ diffusivity of $10^{-4}$ of that in water (Fig. 6), a value comparable to that in another study (7). When $\varepsilon$ is lower ($10^{-5}$), the $O_2$ produced in the early morning cannot quickly diffuse out of the cell, resulting in extremely high intracellular $O_2$ concentrations (about 60 times higher than the far-field ambient $O_2$ concentration) (Fig. 6c). Although not represented in our model, this high intracellular $O_2$ can cause strong oxidative stress (48). When $\varepsilon$ is higher ($10^{-3}$), the modeled cell needs to consume much more carbon in RP, so that the modeled gross fixed C-to-N ratio was substantially increased and the modeled growth rate was greatly decreased, unless the $O_2$ inhibition on $N_2$ fixation was weak (i.e., high $k_{O_2}^{NF}$) (Fig. 6a and b).

The half-saturation coefficient for the $O_2$ inhibition ($k_{O_2}^{NF}$) (equation 3), unknown but probably substantially lower than a typical ambient $O_2$ concentration (0.213 mol $O_2$ m$^{-3}$ at 34 practical salinity units (PSU) and 25°C), is estimated at $10^{-2}$ mol $O_2$ m$^{-3}$. This value of $k_{O_2}^{NF}$, together with an $\varepsilon$ of $10^{-4}$, results in a ratio of modeled gross C-to-N fixations of 49:1 (Fig. 6), which fits well to an observed value of 47:1 from an experiment of *Trichodesmium* (6). By setting a stronger $O_2$ inhibition on $N_2$ fixation, i.e., a lower $k_{O_2}^{NF}$ of $10^{-3}$ mol $O_2$ m$^{-3}$ (see Fig. S5), more carbon is consumed in RP, resulting in a slightly higher ratio of modeled gross fixed C to N (55:1) and a slightly lower growth rate (Fig. 6a and b), while the pattern of the temporal segregation is basically unchanged (Fig. 6d). When the $O_2$ inhibition on $N_2$ fixation is weaker, by setting a higher $k_{O_2}^{NF}$ of $10^{-1}$ mol $O_2$ m$^{-3}$, the model reached a much higher growth rate (0.53 day$^{-1}$) with a lower gross fixed C-to-N ratio (23:1) (Fig. 6a and b). However, in this case, the

modeled intracellular $O_2$ is higher than the ambient $O_2$, even when the $N_2$ fixation rates are high (Fig. 6e), contradictory to our intention of this model and the common understandings that *Trichodesmium* needs to substantially reduce intracellular $O_2$ to allow $N_2$ fixation. The results of this experiment show that the modeled degree of temporal separation depends on the parameter $k_{O_2}^{NF}$, which sets the strength of the $O_2$ inhibition on $N_2$ fixation. The model can resolve a much less pronounced temporal separation (Fig. 6e) than that in the standard model (Fig. 2a) when the $O_2$ inhibition is set to be weaker, while the modeled growth rate of *Trichodesmium* is still comparable to other reported observations (41, 53).

Upon further consideration of other reported observations showing that the gross C:N fixation ratio mainly ranges between 30 and 50 (5, 6, 13, 47), our model sensitivity tests narrowed $\varepsilon$ and $k_{O_2}^{NF}$ estimates to considerably smaller ranges of $\approx 10^{-4.5}$ to $10^{-4}$ and $10^{-2}$ to $10^{-1.5}$ mol $O_2$ $m^{-3}$, respectively, in which the modeled ratios of gross C and N fixations, growth rates, and intracellular levels are likely acceptable (Fig. 6).

Our model also reveals the importance of RP in regulating the intracellular $O_2$ of *Trichodesmium*, in which active RP not only directly consumes $O_2$ but also downregulates PET and thus photosynthetic $O_2$ production (5). Further model experiments without RP showed that the much higher intracellular $O_2$ levels inhibit $N_2$ fixation nearly entirely (see Fig. S6). The active role of RP can also be supported by the observed strong positive correlation between the expression of nitrogenase and cytochrome oxidase (the enzyme of respiration) in *Trichodesmium* (25). The RP of *Trichodesmium* is an extra-high indirect cost of $N_2$ fixation (36) and is a carbon biomass efficiency trade-off strategy commonly adopted by other marine diazotrophs (54–56).

AET can be another important mechanism in $N_2$-fixing *Trichodesmium*. As already discussed, AET partly satisfies the higher ATP demand by $N_2$ fixation. The fraction of PET electrons passing AET is substantially higher in *Trichodesmium* (48% $\pm$ 15%) than in nondiazotrophic cyanobacteria, such as *Synechococcus* (25%) (26, 57). Even the fraction of AET in *Trichodesmium* decreases when it grows with nitrate instead of $N_2$ (8). Our modeled fraction of AET is 39% on a daily basis and has a daily rhythm similar to that of $N_2$ fixation (see Fig. S7b), a phenomenon also found by Milligan et al. (8). Another important role of AET in *Trichodesmium* is to scavenge $O_2$ produced in PSII (27, 28) and to thus protect nitrogenase (58, 59). In our model, AET scavenges 39% of $O_2$ produced in PSII, at rates comparable to those of a previous observation (26). Turning off AET in the model, the increased photosynthetic $O_2$ production (by 56%) elevates RP by 11% and reduces the growth rate by 62%.

There are other possible strategies that *Trichodesmium* may use to manage intracellular $O_2$, but they are not considered in our model. For instance, diazotrophs may dynamically adjust their membrane permeability to $O_2$ by redistributing hopanoid lipids in the membranes to cope with instantaneous requirements (24). The high abundance of *Trichodesmium* found on sinking particles implies that remineralization of particulate organic carbon may create a low-$O_2$ microenvironment for *Trichodesmium* (60). The constitution of *Trichodesmium* colonies may also protect $N_2$ fixation from into-cell $O_2$ diffusion by forming $O_2$-depleted microzones inside the colonies (61). High respiration rates of the heterotrophic bacteria attached to *Trichodesmium* colonies (62) might also help to create a hypoxic microenvironment. However, recent studies found that no anoxia formed inside the colonies during the light period (63, 64). Nevertheless, $N_2$ fixation of *Trichodesmium* colonies is often reported to be lower than that of free trichomes (48, 65, 66). How colony formation helps *Trichodesmium* manage $O_2$ and impacts its $N_2$ fixation, as well as its evolutionary reason, requires further research.

**Conclusions.** In this study, we constructed a physiological model of *Trichodesmium* to explore its conflict in $O_2$-evolving photosynthesis and $O_2$-inhibiting $N_2$ fixation. Our study shows that $N_2$ fixation of *Trichodesmium* is feasible without spatial separation from photosynthesis, consistent with observations in which it occurs in photosynthetic cells. Our model also suggests that the spatial segregation overall may not benefit *Trichodesmium* if substances lose during their transfer across cells. The model provides a mechanistic understanding behind the occurrence of $N_2$ fixation despite the

presence of photosynthesis across the trichome. Proper low cell permeability to $O_2$, respiratory protection, and alternative electron transfer are key processes of *Trichodesmium* in its intracellular management to create the low-$O_2$ window for $N_2$ fixation. Given the diurnal changes of physiological activities simulated (e.g., photosynthetic electron transfer, carbon and nitrogen fixations), our model may be adapted in future studies to provide a further mechanistic insight regarding *Trichodesmium*, for example, into how limiting light intensity and other limiting nutrients such as iron can mediate the ATP and NAPDH production and other processes and then regulate diurnal patterns of growth and $N_2$ fixation. Our model may also be used to advance our understanding of physiological processes in *Trichodesmium* colonies in their dynamic microenvironments by incorporating them into a proper physical framework.

## MATERIALS AND METHODS

In the following, we briefly describe schemes of the model without spatial segregation. The full model description, parameter values, and variables of both models without and with spatial segregation can be found in Text S1 and Tables S1 and S2 in the supplemental material.

**Photosynthetic pathways.** A 12-h daylight cycle is set using a sine function (67) and drives a light-dependent PET rate ($V_{PET}^{I}$, in moles electrons per mole C per second), which is further inhibited by RP as already discussed:

$$V_{PET} = V_{PET}^{I} \cdot e^{-\beta \cdot V_{RP}} \qquad (1)$$

where $V_{RP}$ (in moles C per mole C per second) is the RP rate described later and $\beta$ [in (moles C)$^{-1}$ (moles C · seconds)] is a parameter for the inhibition strength.

The modeled PET is divided into LPET and AET at variable fractions. For each electron through LPET, 0.65 ATP, 0.5 NADPH, and 0.25 $O_2$ are produced, while each electron through AET generates 0.65 ATP but no net NADPH or $O_2$ (27). Note that this ATP production rate by AET is based on a pathway in which electrons cycle through the Mehler reaction (27), which appears to be the dominant AET pathway in *Trichodesmium* (8), although other AET pathways can have different ATP production rates (27).

$N_2$ fixation and C fixation require different ratios of ATP to NADPH (3:1 and 1.9:1, respectively; see below). At each time step, after calculating the $N_2$ fixation rate, the model dynamically adjusts the fraction of AET in PET ($f_{AET}$), and consequently the ratio of produced ATP to NADPH, to fulfill the requirements of the $N_2$ fixation rate and maximize the C fixation (see Text S1 and Fig. S7a). Therefore, our model assumes a fully efficient adjustment of fractioning of PET into LPET and AET.

**$N_2$ fixation.** The $N_2$ fixation, including $N_2$ assimilation to $NH_4^+$ and $NH_4^+$ assimilation to glutamate, in the model consumes 9 ATP and 3 NADPH per fixed N atom (31, 32).

A possible reason that $N_2$ fixation of *Trichodesmium* primarily occurs during the light period is that NADPH required by $N_2$ fixation may be completely provided by PET instead of respiring carbohydrates (39). Therefore, the maximal potential that $N_2$ fixation can reach [$V_{NF}^{max}$, in moles N per (moles C per second)] in our model is when NADPH and ATP produced by PET are fully allocated to $N_2$ fixation:

$$V_{NF}^{max} = V_{PET} \cdot (1 - f_{AET}^{NF}) \cdot \frac{q_{LPET}^{NADPH}}{q_{NF}^{NADPH}} \qquad (2)$$

where $f_{AET}^{NF}$ = 56.7% is the required value of $f_{AET}$ for PET to produce ATP and NADPH at the ratio (3:1) required by $N_2$ fixation, $q_{LPET}^{NADPH}$ = 0.5 mol NADPH (mol electrons)$^{-1}$ is the NADPH production quota of LPET, and $q_{NF}^{NADPH}$ = 3 mol NADPH (mol N)$^{-1}$ is that required by $N_2$ fixation.

$N_2$ fixation in the model also depends on the carbon skeleton (CS; in moles C per mole C), fixed N (in moles N per mole C), and intracellular $O_2$ (in moles $O_2$ per cubic meter):

$$V_{NF} = V_{NF}^{max} \cdot \frac{CS}{CS + k_{CS}^{NF}} \cdot \left(\frac{N_{max} - N}{N_{max}}\right) \cdot \left(1 - \frac{O_2}{O_2 + k_{O_2}^{NF}}\right) \qquad (3)$$

where $k_{CS}^{NF}$ (in moles C per mole C) is the half-saturating coefficient of the carbon skeleton for $N_2$ fixation. We assume that *Trichodesmium* tends to downregulate $N_2$ fixation when the fixed N is approaching maximal N storage ($N_{max}$, in moles N per mole C) (7). The model's $O_2$ inhibition on $N_2$ fixation rate uses a Michaelis-Menten equation (35), in which the value of the half-saturation coefficient ($k_{O_2}^{NF}$) for the inhibition has not been reported in the literature. Model experiments were then conducted to find and pair a $k_{O_2}^{NF}$ value with another parameter, $\varepsilon$, as described below.

**Carbon fixation.** Each inorganic carbon ($C_i$, including $CO_2$ and $HCO_3^-$) is fixed into carbohydrates using 2 NADPH and 3 ATP, based on the stoichiometry of the Calvin-Benson cycle (30). Additional energy of 0.8 ATP per fixed C is used by assuming 50% $C_i$ leakage, 80% $C_i$ from $HCO_3^-$, and a transport cost of 0.5 ATP per $HCO_3^-$ (68, 69). As mentioned above, the rate of carbon fixation is determined with $f_{AET}$ after the $N_2$ fixation rate is calculated.

The carbon skeleton CS value in the model is produced from carbohydrates without energy consumption or carbon loss (37). The production rate of the carbon skeleton ($V_{CS}$, in moles C per mole C per

second) is stimulated by the concentration of carbohydrates ($CH_2O$, in moles C per mole C), as shown using a Michaelis-Menten equation (35) and is inhibited by its own accumulation (7):

$$V_{CS} = v_{CS}^{max} \cdot \frac{CH_2O}{CH_2O + k_{CH_2O}^{CS}} \cdot \frac{CS_{max} - CS}{CS_{max}} \tag{4}$$

where $v_{CS}^{max}$ (in moles C per mole C per second) is the maximal production rate of the carbon skeleton, $k_{CH_2O}^{CS}$ (in moles C per mole C) is the half-saturation constant of carbohydrates for carbon skeleton production, and $CS_{max}$ (in moles C per mole C) is the maximal CS storage.

**Respiratory protection.** To create a low-$O_2$ environment for $N_2$ fixation, high intracellular $O_2$ stimulates RP. The rate of RP is also stimulated by the potential of $N_2$ fixation, which is in turn elevated by light and CS and is limited by fixed N (7, 56). We then parameterized the rate of RP (in moles C per mole C per second), as follows:

$$V_{RP} = v_{RP}^{max} \cdot \frac{O_2}{O_2 + k_{O_2}^{NF}} \cdot (1 - e^{-\alpha_I \cdot I}) \cdot \frac{CS}{CS + k_{CS}^{NF}} \cdot \left(\frac{N_{max} - N}{N_{max}}\right) \tag{5}$$

where $v_{RP}^{max}$ (in moles C per mole C per second) is the maximal RP rate and $\alpha_I$ (per micromole per square meter per second) is the initial slope of the photosynthesis versus light curve.

**$O_2$ diffusion.** The $O_2$ diffusion rate between the cell cytoplasm and ambient environment ($T_{O_2}$, in moles $O_2$ per cubic meter per second) is simulated using a scheme from Staal et al. (40):

$$T_{O_2} = \frac{-2 \cdot \pi \cdot d_{O_2} \cdot L}{V} \cdot \left\{ \frac{1}{\varepsilon} \cdot ln\left(\frac{R}{R + L_g}\right) - ln\left(\frac{R + L_g + L_b}{R + L_g}\right) \right\}^{-1} \cdot (O_2^E - O_2) \tag{6}$$

where $O_2^E$ is the ambient far-field $O_2$ concentration set to a saturating concentration (0.213 mol $O_2$ m$^{-3}$) under typical ocean conditions of 34-PSU salinity and 25°C (70), $d_{O_2}$ (in square meters per second) is the $O_2$ diffusion coefficient in seawater, $L$ (in meters) and $V$ (in cubic meters) are the length and volume of the trichome (simplified to cylindrical geometry), respectively, $\varepsilon$ is the ratio of the $O_2$ diffusion coefficient of the cell membrane to the $d_{O_2}$ and is estimated to be 10$^{-4}$ by model experiments (described below), $R$ (in meters) is the radius of the cytoplasm, $L_g$ (in meters) is the thickness of the cell membrane, and $L_b = 1,024 \cdot (R + L_g)$ is the thickness of the boundary layer (64).

**Integration of state variables during the daytime.** The temporal change rates of state variables of carbohydrates, carbon skeleton, fixed N, and intracellular $O_2$ are represented in ordinary differential equations (ODEs), including all the fluxes described above. Note that NADPH and ATP are not stored but are fully consumed at each time step. Because all the rates described above have been normalized to carbon biomass, either volume, initial biomass, or the biomass concentration of *Trichodesmium* trichome does not need to be included in the model. An exception is for the ODE of intracellular $O_2$ (in moles per cubic meter), in which the cellular carbon biomass concentration ($Q_C$, which is 18,333 mol C m$^{-3}$) (71) is used to convert carbon-normalized biological fluxes of $O_2$:

$$\frac{dO_2}{dt} = (V_{O_2} - V_{O_2}^{RP}) \cdot Q_C + T_{O_2} \tag{7}$$

where ($V_{O_2} - V_{O_2}^{RP}$) is the biological production and consumption of $O_2$ (by RP) in moles of $O_2$ per mole C per second). These ODEs are integrated over the light period by using the MATLAB ode15s integrator (72).

**Biosynthesis and growth rate.** *Trichodesmium* might store fixed C and N during the daytime and assimilate them into biomass, mainly during the dark period (6). Therefore, for simplification, the model calculates the amount of biomass (Bio, in moles C per moles C) that can be synthesized using the carbohydrates, carbon skeletons, and fixed N at the end of the light period. Bio is the smaller of N-based (Bio$_N$) and C-based biomass (Bio$_C$), with Bio$_N$ being calculated by dividing fixed N by the molar ratio N:C (0.159) (46). Bio$_C$ is calculated from the carbohydrates and carbon skeleton, considering the mass and energy balance. The energy needed for biosynthesis is derived from the respiration of carbohydrates ($CH_2O_{BIO}^{RESP}$):

$$Bio_C \cdot q_{BIO}^{ATP} \cdot (1 + \gamma_{MT}) = CH_2O_{BIO}^{RESP} \cdot q_{RESP}^{ATP} \tag{8}$$

where $q_{BIO}^{ATP}$ (= 2 mol ATP per mol C) is the ATP requirement rate for biosynthesis (7), $\gamma_{MT}$ (= 10%) represents additional energy used in maintenance, referring to all cellular processes (e.g., nutrient uptake and DNA protection) that are incalculable but require energy (38), and $q_{RESP}^{ATP}$ (= 5 mol ATP per mol C) is the ATP production rate from respiring carbohydrates (73). Then, the nonrespired carbohydrates and all carbon skeletons can be directly used to synthesize biomass:

$$Bio_C = CH_2O - CH_2O_{BIO}^{RESP} + CS \tag{9}$$

Bio$_C$ and $CH_2O_{BIO}^{RESP}$ are the two unknown variables in equations 8 and 9 and thus can be solved. Noting that all the rates have been normalized to carbon biomass, Bio is therefore the relative increase in biomass over 1 day. The growth rate ($G$) is then the natural log of (1 + Bio) divided by 1 day.

**Optimization of model parameters.** Our optimality-based model assumes *Trichodesmium* can regulate its intracellular processes to maximize its growth (74). In the model without spatial segregation, several important parameters, whose values are largely unknown from the literature, were optimized in large bounded ranges by using the MATLAB global optimizer MultiStart (Table 2). These optimized parameters include those related to

**TABLE 2** Optimized model parameters

| Symbol | Units | Description | Initial range | Value after optimization |
|---|---|---|---|---|
| $k_{CS}^{NF}$ | mol C (mol C)$^{-1}$ | Half-saturating coefficient of carbon skeleton for N$_2$ fixation | $[0, 1]^a$ | 0.06 |
| $v_{CS}^{max}$ | mol C (mol C)$^{-1}$ s$^{-1}$ | Maximal production rate of carbon skeleton | $[0, 5.0 \times 10^{-4}]^b$ | $3.7 \times 10^{-6}$ |
| $k_{CH_2O}^{CS}$ | mol C (mol C)$^{-1}$ | Half-saturating coefficient of carbohydrate for carbon skeleton production | $[0, 1]^a$ | 0.58 |
| $v_{RP}^{max}$ | mol C (mol C)$^{-1}$ s$^{-1}$ | Maximal respiratory protection rate | $[0, 5.0 \times 10^{-4}]^b$ | $4.5 \times 10^{-4}$ (without spatial segregation); $4.0 \times 10^{-4}$ (with spatial segregation) |

$^a$The upper bounds are the maximal potential of organic carbon that can be fixed over the diurnal cycle.
$^b$The upper bounds are the maximal potential of the O$_2$ production rate in photosystem II.

carbon skeleton production ($v_{CS}^{max}$ and $k_{CH_2O}^{CS}$), N$_2$ fixation ($k_{CS}^{NF}$), and RP ($v_{RP}^{max}$). Other parameters (see Table S1 in the supplemental material) are fixed because they are either elemental or are energy stoichiometries of metabolic activities largely constrained by known biochemical reactions, morphological parameters of *Trichodesmium*, and boundary conditions (e.g., light intensity and ambient O$_2$ concentration), or they are derived from model experiments ($k_{O_2}^{NF}$ and $\varepsilon$). To fairly compare results, for the model with spatial segregation we used the same parameter values as those for the model without spatial segregation, except for $v_{RP}^{max}$, which was reoptimized to a lower value in the model with spatial segregation (Table 2), reflecting that RP was less demanded.

**Model experiments with the spatial segregation considering a cost for material transfer.** For the model with spatial segregation, given that N$_2$ fixation is segregated from photosynthesis and is confined in diazocytes, the potential cost for the intercellular materials transfer was considered, including the loss of ATP, NADPH, and carbohydrate transferred into diazocytes and the loss of fixed N transfer into photosynthetic cells. To quantitively evaluate the effect of transfer cost on N$_2$ fixation and growth rates, for simplicity we set the same loss fraction of transferred materials, ranging from 0% to 80%.

**Model availability.** All data generated or analyzed in this study are included in this article and its supplemental material. The code of the model in this study is available on Zenodo (https://doi.org/10.5281/zenodo.6774659).

## SUPPLEMENTAL MATERIAL

Supplemental material is available online only.
**TEXT S1**, PDF file, 0.2 MB.
**FIG S1**, PDF file, 0.1 MB.
**FIG S2**, PDF file, 0.1 MB.
**FIG S3**, PDF file, 1.8 MB.
**FIG S4**, PDF file, 0.1 MB.
**FIG S5**, PDF file, 0.1 MB.
**FIG S6**, PDF file, 0.1 MB.
**FIG S7**, PDF file, 0.6 MB.
**TABLE S1**, PDF file, 0.1 MB.
**TABLE S2**, PDF file, 0.1 MB.

## ACKNOWLEDGMENTS

We thank Ann Pearson at Harvard for her useful suggestions on model optimization and Xiaoli Lu from XMU for her useful support on computation efficiency of the model runs and experiments. This project is partly supported by the National Natural Science Foundation of China (42076153 and 41890802 to Y.-W.L.).

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
