## [Reviewer comments · mSystems]

N₂ fixation in *Trichodesmium* does not require spatial segregation from photosynthesis

Weicheng Luo, Keisuke Inomura, Han Zhang, and Ya-Wei Luo

Corresponding Author(s): Ya-Wei Luo, Xiamen University

Review Timeline:

Submission Date:	June 10, 2022
Editorial Decision:	June 27, 2022
Revision Received:	June 29, 2022
Accepted:	June 30, 2022

Editor: Holly Bik

Reviewer(s): Disclosure of reviewer identity is with reference to reviewer comments included in decision letter(s). The following individuals involved in review of your submission have agreed to reveal their identity: Noelle Held (Reviewer #1)

Transaction Report:

DOI: <https://doi.org/10.1128/msystems.00538-22>

June 27, 2022

Dr. Ya-Wei Luo
Xiamen University
Xiamen
China

Re: mSystems00538-22 (N₂ fixation in *Trichodesmium* does not require spatial segregation from photosynthesis)

Dear Dr. Ya-Wei Luo:

Thank you for submitting your manuscript to mSystems. We have completed our review and I am pleased to inform you that, in principle, we expect to accept it for publication in mSystems. However, acceptance will not be final until you have adequately addressed the reviewer comments.

I am satisfied that the authors have addressed all remaining reviewer concerns (especially reviewer #1's extensive comments), however there are a few minor points to address following an additional round of review (see short comments from two additional reviewers at the end of this email). When submitting your revision, please also ensure all computer code, model parameters, etc. are publicly deposited on GitHub/Zenodo as stated in your data availability statement.

Preparing Revision Guidelines

Sincerely,

Holly Bik

Editor, mSystems

Journals Department
Reviewer comments:

Reviewer #1 (Comments for the Author):

Thank you for thoroughly addressing my prior comments. I am glad to see the sensitivity analyses carried out and that they strengthen the results! I am looking forward to seeing the paper in print.

I'm intrigued by Fig S5 panel b and c -the temporal pattern in the model with low K gives quantitatively similar result in terms of N₂ and C fixed compared to the optimized/standard model, but this time with a much less pronounced temporal separation of N₂ fix and C fix.

I think it is worth expanding the discussion a little bit in lines 331-359 to mention that it is possible for the model to resolve a little temporal separation depending on the k value, while still resolving reasonable albeit not exact biological numbers (such as biomass C:N) relative to experimental/measured data. I would be interested to even see this figure in the main text/methods section such as after the sensitivity analysis figure.

Another specific comment for lines 291-296: "A speculated reason is that those Trichodesmium samples did not grow at its optimal potential, considering that our model is optimized to maximal growth rate."

I'm not sure this is a valid argument because in both cited papers the cultures were growing at a rate closer to the maximum growth rate suggested in the model (~0.25 per day in the model, 0.7 per day in Held et al and 0.3-0.5 per day than that in Cai and Gao) than is usually achieved in experimental systems. Some experiments observe temporal separation in systems with much slower growth rates (such as 0.1 per day) than these.

Overall I am impressed by the revised paper and congratulate the authors.

Reviewer #3 (Comments for the Author):

I do not feel that the authors have sufficiently addressed the detailed comments from reviewer #. It has also long been proposed that Trichodesmium creates a micro-climate of low oxygen concentration by having the filaments create colonies, and having active nitrogen fixation in the inner most cells of the colonies. This is not addressed by the authors.

While the model proposed in an attractive one, I am not confident that sufficient data, analysis and explanation is presented in this manuscript to overturn the current model. This model would be strengthened by having actual measurable data. I understand how difficult this is to perform, but it would be the best supporting evidence for this model.

Luo et al. Re-review

Thank you to the authors for thoroughly addressing my prior comments. I am glad to see the sensitivity analyses carried out and that they strengthen the results! I am looking forward to seeing the paper in print.

I'm intrigued by Fig S5 panel b and c –the temporal pattern in the model with low K gives quantitatively similar result in terms of N₂ and C fixed compared to the optimized/standard model, but this time with a much less pronounced temporal separation of N₂ fix and C fix. I think it is worth expanding the discussion a little bit in lines 331-359 to mention that it is possible for the model to resolve a little temporal separation depending on the k value, while still resolving reasonable albeit not exact biological numbers (such as biomass C:N) relative to experimental/measured data. I would be interested to even see this figure in the main text/methods section such as after the sensitivity analysis figure.

Another specific comment for lines 291-296: "A speculated reason is that those Trichodesmium samples did not grow at its optimal potential, considering that our model is optimized to maximal growth rate."

I'm not sure this is a valid argument because in both cited papers the cultures were growing at a rate closer to the maximum growth rate suggested in the model (~0.25 per day in the model, 0.7 per day in Held et al and 0.3-0.5 per day than that in Cai and Gao) than is usually achieved in experimental systems. Some experiments observe temporal separation in systems with much slower growth rates (such as 0.1 per day) than these.

Overall I am impressed by the revised paper and congratulate the authors.

We have taken all the comments of the reviewers into account in the revision; replies to each of the comments are provided below in blue fonts. Please note that all the line numbers mentioned in the response refer to the Marked-up Manuscript.

Reviewer comments:

Reviewer #1 (Comments for the Author):

Thank you for thoroughly addressing my prior comments. I am glad to see the sensitivity analyses carried out and that they strengthen the results! I am looking forward to seeing the paper in print.

Response:

We are grateful to the Reviewer for the positive comments.

I'm intrigued by Fig S5 panel b and c -the temporal pattern in the model with low K gives quantitatively similar result in terms of N₂ and C fixed compared to the optimized/standard model, but this time with a much less pronounced temporal separation of N₂ fix and C fix.

I think it is worth expanding the discussion a little bit in lines 331-359 to mention that it is possible for the model to resolve a little temporal separation depending on the k value, while still resolving reasonable albeit not exact biological numbers (such as biomass C:N) relative to experimental/measured data. I would be interested to even see this figure in the main text/methods section such as after the sensitivity analysis figure.

Response:

We thank the Reviewer for the comments. As suggested by the reviewer, we have moved Fig. S5b and S5c to the main Figure 6 (the sensitivity analyses). We also have added the discussion in **Lines 343-348 (Page 17):**

“The results of this experiment show that the modeled degree of temporal separation depends on the parameter $k_{O_2}^{NF}$ that sets the strength of the O₂ inhibition on N₂ fixation. The model can resolve a much less pronounced temporal separation (Fig. 6e) than that in the standard model (Fig. 2a) when the O₂ inhibition is set to be weaker, while the modeled growth rate of *Trichodesmium* is still comparable to some observations (e.g., 41, 54).”

Another specific comment for lines 291-296: "A speculated reason is that those *Trichodesmium* samples did not grow at its optimal potential, considering that our model is optimized to maximal growth rate."

I'm not sure this is a valid argument because in both cited papers the cultures were growing at a

rate closer to the maximum growth rate suggested in the model (~0.25 per day in the model, 0.7 per day in Held et al and 0.3-0.5 per day than that in Cai and Gao) than is usually achieved in experimental systems. Some experiments observe temporal separation in systems with much slower growth rates (such as 0.1 per day) than these.

Response:

We agree with the Reviewer that the statement might be not valid. Therefore, we have deleted the sentence “A speculated reason is that those *Trichodesmium* samples did not grow at its optimal potential, considering that our model is optimized to maximal growth rate.”

Overall I am impressed by the revised paper and congratulate the authors.

Response:

We are thankful for the Reviewer’s comment.

Reviewer #3 (Comments for the Author):

I do not feel that the authors have sufficiently addressed the detailed comments from reviewer #. It has also long been proposed that *Trichodesmium* creates a micro-climate of low oxygen concentration by having the filaments create colonies, and having active nitrogen fixation in the inner most cells of the colonies. This is not addressed by the authors.

While the model proposed in an attractive one, I am not confident that sufficient data, analysis and explanation is presented in this manuscript to overturn the current model. This model would be strengthened by having actual measurable data. I understand how difficult this is to perform, but it would be the best supporting evidence for this model.

Response:

We thank the Reviewer for the comments. We have been aware of the hypothesized low-oxygen micro-environment in centers of *Trichodesmium* colonies to potentially support N₂ fixation, and have discussed this in the last paragraph of the Discussion (**Lines 377-389, Pages 18-19**). Interestingly, however, as we have discussed there, recent studies found that oxygen inside *Trichodesmium* colonies is greatly elevated during the light period (Eichner et al., 2017; Klawonn et al., 2020).

We understand the reviewer’s concern that sufficient data and analyses are needed to overturn an existing model by a new model. However, as have been discussed in our manuscript and supported by another reviewer, the concept of the spatial segregation in *Trichodesmium* has been under debate. In the manuscript, we have compared some of our model results with historical observations. The reviewer is certainly right that more measurable data will further support our model, which, however, is beyond the scope of this study.

References:

Eichner MJ, Klawonn I, Wilson ST, Littmann S, Whitehouse MJ, Church MJ, Kuypers MM, Karl DM, Ploug H. 2017. Chemical microenvironments and single-cell carbon and nitrogen uptake in field-collected colonies of *Trichodesmium* under different $p\text{CO}_2$. ISME J 11:1305-1317.

Klawonn I, Eichner MJ, Wilson ST, Moradi N, Thamdrup B, Kummel S, Gehre M, Khalili A, Grossart HP, Karl DM, Ploug H. 2020. Distinct nitrogen cycling and steep chemical gradients in *Trichodesmium* colonies. ISME J 14:399-412.

June 30, 2022

Dr. Ya-Wei Luo
Xiamen University
Xiamen
China

Re: mSystems00538-22R1 (N₂ fixation in *Trichodesmium* does not require spatial segregation from photosynthesis)

Dear Dr. Ya-Wei Luo:

I am satisfied that the authors have addressed all remaining reviewer concerns, and I am now happy to recommend final acceptance for this manuscript.

Your manuscript has been accepted, and I am forwarding it to the ASM Journals Department for publication. For your reference, ASM Journals' address is given below. Before it can be scheduled for publication, your manuscript will be checked by the mSystems production staff to make sure that all elements meet the technical requirements for publication. They will contact you if anything needs to be revised before copyediting and production can begin. Otherwise, you will be notified when your proofs are ready to be viewed.

Publication Fees:

If you would like to submit a potential Featured Image, please email a file and a short legend to mSystems@asmusa.org. Please note that we can only consider images that (i) the authors created or own and (ii) have not been previously published. By submitting, you agree that the image can be used under the same terms as the published article. File requirements: square dimensions (4" x 4"), 300 dpi resolution, RGB colorspace, TIF file format.

We recognize that the video files can become quite large, and so to avoid quality loss ASM suggests sending the video file via <https://www.wetransfer.com/>. When you have a final version of the video and the still ready to share, please send it to mSystems staff at mSystems@asmusa.org.

Sincerely,

Holly Bik
Editor, mSystems

Journals Department
Fig. S4: Accept
Text S1: Accept
Fig. S1: Accept
Table S1: Accept
Fig. S3: Accept
Table S2: Accept
Fig. S5: Accept
Fig. S6: Accept
Fig. S7: Accept
Fig. S2: Accept